# Attitudes toward the Legalization of Euthanasia or Physician-Assisted Suicide in South Korea: A Cross-Sectional Survey

**DOI:** 10.3390/ijerph19095183

**Published:** 2022-04-24

**Authors:** Young Ho Yun, Jin-Ah Sim, Yeani Choi, Hyejeong Yoon

**Affiliations:** 1Department of Human Systems Medicine, Seoul National University College of Medicine, Seoul 03080, Korea; xwwjdx@snu.ac.kr (Y.C.); yoonhj5@snu.ac.kr (H.Y.); 2Department of Family Medicine, Seoul National University College of Medicine, Seoul 03080, Korea; 3School of AI Convergence, Hallym University, Chuncheon 200160, Korea; jin-ah.sim@hallym.ac.kr; 4Department of Health Policy and Management, Seoul National University College of Medicine, Seoul 03080, Korea; 5College of Liberal Studies, Seoul National University, Seoul 03080, Korea

**Keywords:** euthanasia, physician-assisted suicide, legalization, attitudes, general population

## Abstract

This study aimed to investigate the general South Korean public attitudes toward the legalization of euthanasia or physician-assisted suicide (EAS) and examine the reasons underpinning these attitudes. From March–April 2021, we conducted a cross-sectional survey of a representative national sample of 1000 South Koreans aged 19 years or older. Three in four participants (76.4%) expressed positive attitudes toward the legalization of EAS. Participants who agreed with this legalization reported “meaninglessness of the rest of life” and “right to a good death” as their main reasons. Participants who disagreed with the legalization of EAS reported “respect for life”, “violation of the right to self-determination”, “risk of abuse or overuse”, and “violation of human rights” as theirs. In the multivariate logistic regression analyses, participants with poor physical status (adjusted odds ratio [aOR]: 1.41, 95%; confidence interval [CI]: 1.02–1.93) or comorbidity (aOR: 1.84, 95%; CI: 1.19–2.83) showed positive attitudes toward the legalization of EAS. In summary, most of the general South Korean population regards the legalization of EAS positively, especially participants with poor physical status or comorbidity.

## 1. Introduction

Euthanasia and physician-assisted suicide (EAS) is legal in only a few countries [1,2,3,4,5,6,7] and a fiercely debated issue, particularly in relation to the fact that it is a basic human right worldwide [6,8,9,10,11,12,13]. Nevertheless, similar right-to-die laws are currently introduced in other jurisdictions [14].

Recent systemic reviews showed that most of the studies on this matter investigated attitudes toward EAS and that age, religion, education, and socio-economic status were consistent predictors of these attitudes [15].

Recently, reports on EAS-related cases—such as the physician-assisted suicide (PAS) of Dr. David Goodall, a 104-year-old Australian scientist in Switzerland—have dramatically increased, and EAS is gaining public attention as a human rights issue [16,17].

Dr. Goodall did not suffer from any serious illness. However, he did not want to live longer due to his diminishing independence. Therefore, he traveled to Switzerland, where PAS is allowed, and died in May 2018 via this procedure.

In South Korea, EAS is illegal and considered a crime against life, but it is also a controversial topic subject to a lively bioethical debate [6,12,16,18]. A Korean Supreme Court case that ordered physicians to remove an elderly woman in a persistent vegetative state from a ventilator in 2009 led to the enactment of the Act on Decisions on Life-Sustaining Treatment for Patients at the End-of-Life in 2016 [6,19,20]. Nonetheless, EAS still cannot be performed in Korea as it remains prohibited by the law.

Public attitudes toward EAS have not been extensively researched in South Korea, and existing studies do not focus on issues stimulated by the current worldwide debate [6,19]. In a 2008 survey, 3840 individuals—patients, family caregivers, the general population, and physicians—from Korea showed that about 50% of those in the patient and general population groups supported EAS, compared to less than 40% of family caregivers and less than 10% of physicians [20]. In a 2016 survey, four groups showed that about 30–40% of those in the patient and general population groups supported EAS, compared to about 20–30% of family caregivers and physicians [6].

Therefore, given the lack of research on this topic and the vignette regarding Dr. David Goodall’s PAS, we investigate the general South Korean public attitudes toward EAS and the reasons underpinning these attitudes [21].

## 2. Materials and Methods

### 2.1. Study Design, Setting, and Population

We aimed to recruit 1000 members of the general population from 17 main provinces of the Republic of Korea. The participants had to be 19 years or older and able to understand the objectives of the survey. Those who could not speak, hear or read Korean or had difficulty in understanding the contents of the questionnaire due to vision or hearing problems were excluded. Based on guidelines provided by the 2020 Korean population census data, the survey was conducted in each district, considering age and sex. We used the probability-proportional-to-size sampling technique to obtain a representative national sample [22].

The survey was conducted from March–April 2021. The interviewer visited the home or workplace of each individual selected for the sample to conduct an eligibility evaluation. We selected 1800 eligible respondents to account for lower participation rates. Among them, 1000 individuals who had a strong understanding of the survey’s purpose and method were recruited (response rate was 55.6%). Additionally, they responded to the self-report questionnaire in the presence of the interviewer, which allowed the interviewer to provide detailed explanations regarding the study, while also allowing for privacy and anonymity.

This study was reviewed and approved by the institutional review board (IRB) of the Seoul National University Hospital as an IRB Review Exemption study since we collected survey questionnaire data from unspecified participants. Additionally, these data did not include personally identifiable or sensitive information (IRB No. 2102-098-1197).

### 2.2. Measurement

We constructed a questionnaire to examine participants’ attitudes toward the legalization of EAS and their reasons for agreeing or disagreeing with this legalization. The attitudes of each participant were assessed in the following context: “In May 2018, 104-year-old Australian ecologist Dr. David Goodall died by physician-assisted suicide in Switzerland. Two Koreans have already been allowed to die in the same manner, while 107 Koreans are members of a physician-assisted suicide group. The Netherlands, Luxembourg, Belgium, Switzerland, Canada, Australia, and eight states in the US allow for euthanasia or physician-assisted suicide. In Korea too, there is an argument for legislating physician-assisted suicide or euthanasia. What do you think about physician-assisted suicide or euthanasia?” The responses were later assessed based on the following Likert scale: “Very much agree” (1), “agree” (2), “disagree” (3), and “very much disagree” (4). The reasons for agreeing or disagreeing with the legalization of EAS were collected.

Moreover, we collected sociodemographic variables, including age, gender, education level, income level, religion, comorbidity, health status, and political orientation through the survey. Health status was measured as perceived health status on a 5-point scale: “Excellent” (1), “very good” (2), “good” (3), “poor” (4), and “bad” (5) [23]. Political orientation was assessed on a 5-point scale: “Very liberal” (1), “somewhat liberal” (2), “middle-of-the-road” (3), “somewhat conservative” (4), and “very conservative” (5) [24].

### 2.3. Statistical Analyses

We used the G-power program to set the appropriate sample size in order for the default setting to remain on the effect size as (0.2), α (0.05), and 1-β (0.95). Considering gender (2, male, female), age (5, 20s, 30s, 40s, 50s, 60s or older), and regional size (4, special, wide area, city, county), the number of groups with the appropriate sample size was 1000.

We collected participants’ sociodemographic characteristics, and then asked them to complete the questionnaire regarding their attitudes toward the legalization of EAS. Moreover, we collected data on their reasons for agreeing or disagreeing with this legalization. All of the data were collected anonymously, and the descriptive analyses were conducted by a researcher who was not involved in data collection.

We used univariate logistic regression analyses to identify factors related to attitudes toward EAS legalization. The univariate logistic regression models adopted here examined the association between attitudes toward EAS and sociodemographic characteristics, political orientation, and health-related components, including comorbidity and health status. Moreover, with age, sex, education, income, marital status, religion, region, housing, job status, and disease status, we used multivariate logistic regression analyses to evaluate the association between poor physical health status, which was a significant factor in the univariate analyses and agreement with EAS. We performed the statistical analyses using SAS statistical software (version 9.4; SAS Institute, Cary, NC, USA). Statistical significance was defined as a two-sided *p*-value below 0.05.

## 3. Results

Table 1 presents participants’ sociodemographic characteristics, including their health status. The mean (SD) age was 48.0 years (19.7 years). Figure 1 shows participants’ attitudes toward the legalization of EAS. Three in four participants (76.4%) held positive attitudes toward the legalization of EAS. Table 2 shows the attitudes toward EAS, in accordance with the intersection of sex × age.

Table 3 shows participants’ reasons for agreeing or disagreeing with the legalization of EAS. The group that supported the legalization of EAS selected “the meaninglessness of the rest of life” and “the right to a good death” as their main rationales. The group that disagreed with the legalization of EAS selected “respect for life”, “violation of the right to self-determination”, “risk of abuse or overuse”, and “violation of human rights protection” as theirs.

The univariate logistic regression analyses of factors related to attitudes toward the legalization of EAS showed that data, age, sex, level of education, income, religion, comorbidity, and political orientation did not significantly influence attitudes toward EAS, whereas physical health status was found to be a significant factor in affecting attitudes toward EAS (Appendix A).

The multivariate logistic regression analyses of the correlation of physical status, age, sex, education, income, marital status, religion, region, housing, job status, and disease status with attitudes toward the legalization of EAS showed that participants with poor physical status or comorbidity had positive attitudes toward the legalization of EAS (Table 4).

## 4. Discussion

This study investigated the general South Korean public attitudes toward EAS and their reasons for these attitudes.

Seventy-six percent of the participants agreed with the legalization of EAS in the country, which indicates a substantial growing public support for EAS, similar to those found in countries, such as England (75.8%) [10] and Switzerland (81.7%) [21].

Our results suggest that a discussion regarding the legalization of EAS may be necessary for Korea [25]. Prior research leads to similar conclusions. A study investigating cross-country differences in attitudes toward euthanasia showed that the residents of 23 of 24 high-income countries view euthanasia as more justifiable [25]. Globally, the number of reported euthanasia deaths has increased annually, specifically among older individuals [26,27].

Previous studies have found that respect for autonomy and preferences for control over the end of one’s life are positively associated with support for EAS [25,28,29]. In contrast, this study found that the main reasons for positive attitudes toward EAS are “the meaninglessness of the rest of life”, “the right to a good death”, “relief from suffering”, and “not being a burden”. These reasons support the legalization of EAS, but may lead to considerable debate, especially regarding EAS for healthy older people, despite their autonomous decision-making. In turn, this debate may create barriers to the legalization of EAS practices [28,30,31,32].

Previous studies have identified age, education, income, religion, political orientation, self-rated health, and the availability of voluntary workers as potential factors associated with attitudes toward EAS [7,21,33,34,35]. Our multivariate logistic regression analyses identified poor health and comorbidity as factors associated with attitudes toward EAS. Recent systemic reviews of older adults’ attitudes toward EAS showed that younger age, higher education, higher socio-economic status, and lower religiousness were the most consistent predictors of attitudes toward EAS [15]. However, findings from the reviews also indicated difficulty in comparing various studies due to the differing participant characteristics and outcome measures used therein [15]. This study suggests that as people age, they develop comorbidities and their health deteriorates, and an aging population with comorbidities and poor health might expect an easy death through EAS. These findings are expected and understandable. However, further research regarding demographic factors’ influence on attitudes toward EAS is required [36].

The findings of this study can deepen our understanding of the public normative reasons regarding EAS and will help in informing future discussions on its legalization [37]. Notably, a previous study in Switzerland found a positive association between trust in the legal system and support for the legalization of physician-assisted suicide. This finding has important implications for the legalization of EAS in Korea, as well [28].

Additionally, the public opinions regarding EAS confirmed by this study must be further accounted for in discourse and policymaking regarding patient autonomy, care for dignity, and quality palliative care [38]. In Korea, the provision of palliative care is currently suboptimal and mostly restricted to cancer patients. Therefore, the negative impact of the legalization of EAS on the development of palliative care is a serious concern [38]. Without the effective right to quality palliative care, the legalization of EAS can pose problems, such as threatening messages to vulnerable individuals that cause them to believe they are burdening their families or society, hasty acceptance without the support of advisory entities or the potential impairment of the development of palliative care [29,39,40].

This study finds that the general population considers advanced care planning and palliative care as alternatives to EAS. However, in accordance with another relevant study by Gerson, no clear and uniform relationship between palliative care and EAS has been found in various locations, such as Quebec (Canada), Flanders (Belgium), and Oregon (USA) [41].

Furthermore, palliative sedation should not be provided in response to requests for EAS in standard palliative care [11]. However, when suffering is refractory to palliative care, some patients with terminal illness, who have a strong desire to maximize their autonomy and dignity, request EAS. Therefore, EAS may be a possible last resort in conjunction with improved palliative care.

Although an EAS law might contain strict safeguards, such as prognostic requirements and psychiatric exclusion, a broad definition of unbearable suffering, unpunished non-reporting, and expansion of EAS to children, people with mental illness, and dementia could limit these safeguards for vulnerable patients [42]. An individual autonomous choice for EAS is shaped by social norms and constrained by feasibility, and as a result, a slippery slope may emerge [42,43]. Whereas unbearable suffering at the end-of-life is a common fear, people should know that they can expect relief from suffering due to medical advances in palliative care [42]. Additionally, healthcare providers should be aware of the complexity of the EAS issue and provide appropriate support and resources to patients in their decision-making process [15].

We encountered certain limitations that should be considered when interpreting this study. First, the results of this study are likely driven by Dr. David Goodall’s PAS and the impact of the COVID-19 pandemic. Therefore, further studies should be conducted to confirm their independence from factors related to the timing of this study. Second, the definition of EAS may differ across various surveys in different countries, which is important to note when comparing attitudes toward EAS across countries. However, we defined EAS using common words and phrases, including a description of the PAS of Dr. David Goodall in Switzerland. Furthermore, we confirmed the participants’ attitudes and the reasons for their positive or negative attitudes toward EAS. Third, Dr. Goodall’s PAS was reported in the media and turned into a big social issue in Korea, and was thus used in the survey to understand EAS realistically. Since the prompt focuses on a non-Korean man who traveled to Switzerland for PAS, the cultural context of the case prompt might have been completely foreign to South Korean respondents. The context of EAS, that is, how EAS questions are framed, makes a difference in how participants respond. Additionally, a single-item attitude measure might not correctly gauge the attitudes of people toward EAS. Future studies should adopt common and explicit definitions of EAS that allow for better consideration of personal, social, and cultural factors on attitudes toward EAS [15]. Fourth, although participants completed the self-reported questionnaire, the presence of the interviewer might have not allowed for privacy and anonymity. Fifth, with a response rate of only 55.6%, findings from this study’s sample population may not be representative of the general population due to non-response bias. Sixth, we could not perform inverse probability weighting techniques due to the non-responders’ lack of information, and therefore, a concern of selection bias remains. However, the sociodemographic characteristics of the survey participants included in this study (N = 1000) were similar to those of the Korean population with regards to age (20–29 years: 15.3%, 30–39 years: 15.5%, 40–49 years: 18.9%, 50–59 years: 20.0%, 60–69 years: 16.7%, ≥70 years: 13.6% in the Korean population) and sex (men: 49.9%, women: 50.1% in the Korean population), suggesting a low possibility of selection bias. Finally, we surveyed a small group of individuals who were over 70 years of age and a large group who were under 50 years of age.

## 5. Conclusions

Most of the survey participants agreed with the legalization of EAS, especially participants with poor physical status or comorbidity, suggesting that discussions regarding this matter may commence in Korea in the near future.

## Figures and Tables

**Figure 1 ijerph-19-05183-f001:**
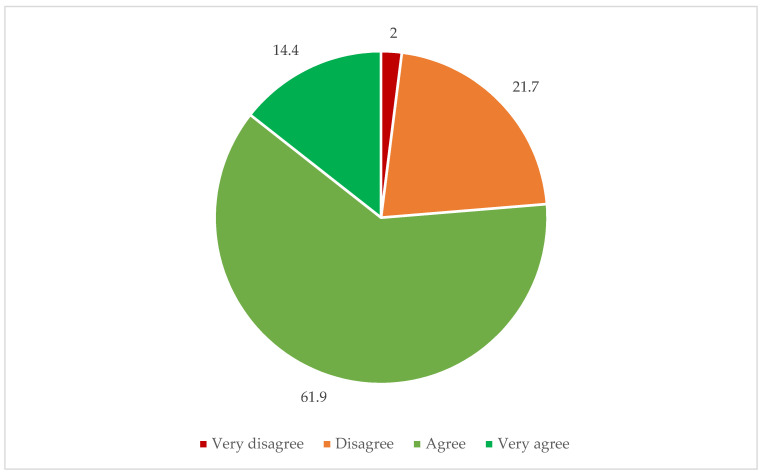
Participants’ attitudes toward the legalization of euthanasia and physician-assisted suicide (%).

**Table 1 ijerph-19-05183-t001:** Participants’ sociodemographic characteristics, including their health status.

Variable	Description	N	%
		Mean	SD
Age		47.96	14.66
		N	%
Age	20–29	166	16.60
	30–39	166	16.60
	40–49	205	20.50
	50–59	209	20.90
	60–69	164	16.40
	≥70	90	9.00
Sex	Male	503	50.30
	Female	497	49.70
Comorbidity	No	734	73.40
	Hypertension		
	Dyslipidemia		
	Diabetes mellitus	70	7.00
	Musculoskeletal disease	24	2.40
	Liver disease	10	1.00
	Others	26	2.60
Education	College graduate	541	54.10
	High school graduate	361	36.10
	Middle school or less	90	9.80
Income	≥5000	276	27.60
(1000 Won)	4000–5000	274	27.50
	3000–4000	228	22.80
	<3000	221	22.10
Marriage	Married	714	71.40
	Not married	286	28.60
Residence	Urban	460	46.00
	Rural/suburban	540	54.00
Religion	Religious	360	36.00
	Non-religious	640	64.00
Job status	Occupied	747	74.70
	Non-occupied	253	25.30

**Table 2 ijerph-19-05183-t002:** Participants’ attitudes toward the legalization of euthanasia and physician-assisted suicide (N = 1000).

Male, N (%)		
Age	20–49	≥50
Agree	201 (40.0)	176 (35.0)
Disagree	76 (15.1)	50 (9.9)
Female, N (%)		
Age	20–49	≥50
Agree	200 (40.2)	186 (37.4)
Disagree	60 (12.1)	51 (10.3)

**Table 3 ijerph-19-05183-t003:** Respondents’ reasons for agreeing or disagreeing with the legalization of EAS.

	N (%)
Reasons for agreement (N = 763)	
Meaninglessness of the rest of life	235 (30.8)
Right to a good death	198 (26.0)
Alleviation of suffering	157 (20.6)
Family suffering and burden	113 (14.8)
Social burden due to medical expenses and care	35 (4.6)
No violation of human rights	27 (3.1)
Importance of the right to self-determination	1 (0.1)
Reasons for disagreement (N = 237)	
Respect for life	105 (44.3)
Violation of the right to self-determination	37 (15.6)
Risk of abuse or overuse	31 (13.1)
Violation of human rights	29 (12.2)
Risk of misdiagnosis	23 (9.7)
Possibility of recovery	12 (5.1)

EAS: Euthanasia and physician-assisted suicide.

**Table 4 ijerph-19-05183-t004:** Multivariate logistic regression analyses of factors related to the legalization of EAS.

Factors	Agree vs. Disagree (Ref)	
	aOR	95% CI
Age in years at survey		
<50 (Ref)	NS	
≥50		
Sex		
Male (Ref)	NS	
Female		
Comorbidity		
None (Ref)	1	
More than one	1.835	1.189–2.832
Educational background		
College graduate or post-graduate (Ref)	NS	
HS graduate/GED or below		
Monthly household incomes		
≥$3000 (Ref)	NS	
<$3000		
Marital status		
Single/widowed/divorced/separated (Ref)	NS	
Married/living with a partner		
Religion		
Religious (Ref)	NS	
Non-religious		
Rural/Urban area		
Rural/suburban (Ref)	NS	
Urban		
Political Tendency		
Center (Ref)	NS	
Progressive		
Conservative		
Job status		
Occupied (Ref)	NS	
Non-occupied		
Physical Health Status		
≥Very Good (Ref)	1	
<Very Good	1.405	1.023–1.930

EAS: Euthanasia and physician-assisted suicide; Ref: Reference; aOR: Adjusted odds ratio; CI: Confidence interval; NS: Non-significant.

## Data Availability

Data are available upon reasonable request.

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
