# Peer review of "Attitudes toward the Legalization of Euthanasia or Physician-Assisted Suicide in South Korea: A Cross-Sectional Survey"

_ijerph, 2022, doi:10.3390/ijerph19095183_

Round 1
Reviewer 1 Report
This study aimed to investigate the general South Korean public's attitudes toward the legalization of euthanasia or physician-assisted suicide (EAS) and examine the reasons of these attitudes. Although the study is interesting and timely, the writing is somewhere superficial and the methodological part should be re-written trying to add much more details.
INTRODUCTION
Please, may you better explain Dr. David Goodall case?
Authors stated that “Public attitudes toward EAS are relatively infrequently studied”. This is not true since there are a number of studies focused on this topic. Please, better revise the literature.
MATERIALS AND METHODS
Why you aimed to recruit 1,000 members? What about power analysis?
How the sample has been selected?
Please indicate inclusion and exclusion criteria.
Please explain what do you mean with “55.6% valid response rate”. What about the features of the rest of the sample?
Authors stated that they “constructed a questionnaire to examine attitudes toward the legalization of EAS and the reasons for these attitudes”. Much more details should be added. How the questionnaire has been constructed?
DISCUSSION
Results may be better discussed and methodological limitations must be added.
Minor remarks:
Line 143: please correct “. [24[ Prior”.
Author Response
Response to 1st Reviewer
Comment #1. Better explain Dr. David Goodall case in INTRODUCTION.
As requested by the reviewer, we have added more details about Dr. David Goodall’s case in INTRODUCTION as follows (page 2. Lines 35-43):
Recent systemic reviews of older adults’ requests for or attitudes toward EAS showed that most studies investigated attitudes toward EAS, and that age, religiosity, education, and socio-economic status were consistent predictors of such attitudes [15].
Recently, reports about cases involving EAS, such as the case of physician-assisted suicide (PAS) of 104-year-old Australian scientist Dr. David Goodall in Switzerland, have dramatically increased, and EAS is gaining public attention as a human rights issue [16,17].
Dr. Goodall did not suffer from any serious illness; however, he did not want to live longer due to his diminishing independence. He thus traveled to Switzerland, where PAS is allowed, and died by PAS in May 2018.
Comment #2. A number of studies focused on public attitudes toward EAS
With regard to the reviewer's comment, we revised the sentence as follows (page 2-3, lines 50-52):
Public attitudes toward EAS have not been extensively researched in South Korea, and existing studies do not focus on issues stimulated by the current worldwide debate [6,19].
Comment #3. Sample size recruiting 1,000 members and power analysis
We have added more information about the appropriate sample size and power analysis in the Statistical Analysis sub-section under MATERIALS AND METHOD as follows (page 4, lines 105-108):
We used Gpower to set the appropriate sample size to keep the default setting for effect size (0.2), α (0.05), and 1-β (0.95). The number of groups was gender (2, male, female), age (5, 20's, 30's, 40's, 50's, 60's or older), and regional size (4, special, wide area, city, county) = 40. Therefore, the appropriate sample size was 1,000.
Comment #4. More information about the inclusion and exclusion criteria
We have added more information about the inclusion and exclusion criteria in the study methods as follows (page 3, lines 65-68):
The participants had to be 19 years or older and be able to understand the objectives of the survey to be included. Those who could not speak, hear, or read Korean and had difficulty in understanding the contents of the questionnaire due to vision and hearing problems were excluded.
Comment #5-1. 55.6% valid response rate
We have added more explanation in the study methods as follows (page 3, lines 74-77):
We chose 1,800 eligible respondents to account for lower participation rates. Among them, 1,000 individuals who had a strong understanding of the survey’s purpose and method were finally recruited (response rate was 55.6%) and responded to the self-reported questionnaire in the presence of the interviewer.
Comment #5-2. The features of the rest of the sample
As rightly pointed out by the reviewer, we did not investigate the features of the rest of the sample, and have added this point as a limitation in the DISCUSSION section as follows (page 10, lines 248-254):
Sixth, we could not perform inverse probability weighting techniques due to a lack of information of non-responders and, therefore, a concern of selection bias remains; the sociodemographic characteristics of the survey participants included in this study (n=1,000) were similar to those of the Korean population with regard to age (20–29 years: 15.3%, 30–39 years: 15.5%, 40–49 years: 18.9%, 50–59 years: 20.0%, 60–69 years: 16.7%, ≥70 years: 13.6% in the Korean population) and sex (men: 49.9%, women: 50.1% in the Korean population), suggesting low possibility of selection bias.
Comment #6. Much more details about construction of a questionnaire to examine attitudes toward the legalization of EAS and the reasons for these attitudes
We have added more details about the construction of the questionnaire to examine attitudes toward the legalization of EAS and the reasons for these attitudes in METHODS as follows (page 3, lines 84-96):
We constructed a questionnaire to examine attitudes toward the legalization of EAS and the reasons for agreeing or disagreeing with its legalization. Each participant’s attitudes toward the legalization of EAS were assessed in the following context: “In May 2018, 104-year-old Australian ecologist Dr. David Goodall died by physician-assisted suicide in Switzerland. Two Koreans have already been allowed to die by physician-assisted suicide, while 107 Koreans are members of a physician-assisted suicide group. The Netherlands, Luxembourg, Belgium, Switzerland, Canada, Australia, and eight states in the US allow euthanasia or physician-assisted suicide. In Korea too, there is an argument for legislating physician-assisted suicide or euthanasia. What do you think about physician-assisted suicide or euthanasia?” The responses were later assessed based on the following Likert scale: “very much agree” (1), “agree” (2), “disagree” (3), and “very much disagree” (4). The reasons for agreeing or disagreeing with the legalization of EAS are shown in Table 3.
Comment #7. Better discussion and methodological limitations
We have added more details about the discussion and methodological limitations as follows:
DISCUSSION (pages 9, lines 186-192)
Recent systemic reviews of older adults’ attitudes toward EAS showed that younger age, higher education, higher socio-economic status, and lower religiosity were the most consistent predictors of attitude toward EAS [15]. However, findings from the reviews also indicated that there was difficulty in comparison across studies because of the variety of participants’ characteristics and outcome measures [15]. This study suggests that as people age, they develop comorbidities and their health deteriorates, and an aging population with comorbidities and poor health might expect easy death through EAS.
DISCUSSION (page 9-10, lines 218-227)
Although an EAS law might contain strict safeguards such as prognostic require-ments and psychiatric exclusion, a broad definition of unbearable suffering, unpunished non-reporting, and expansion of EAS to children, people with mental illness, and dementia could limit the safeguards for vulnerable patients [42]. An individual autonomous choice for EAS is shaped by social norms and constrained by feasibility, and as a result, a slippery slope can be real [42,43]. Whereas unbearable suffering at the end of life is a common fear; people should know that they would be able to expect relief from suffering as a result of medical advances in palliative care [42]. Additionally, healthcare providers should be aware of the complexity of the EAS issue, and provide appropriate support and resources to patients in their decision-making process [15].
Methodological limitations (pages 10, lines 236-256)
Third, the PAS of Dr. Goodall was reported in the media and became a big social issue in Korea, and this case was used in the survey to realistically understand EAS. As the prompt focuses on a man who is not Korean and who goes to Switzerland to die of PAS, the cultural context of the case prompt might be completely foreign to South Korean respondents. The context of EAS, that is, how EAS questions are framed, makes a difference in how participants respond. Additionally, a single-item attitude measure might not correctly gauge the attitudes of people toward EAS. Future studies should adopt common and explicit definitions of EAS that allow for better consideration of personal, social, and cultural factors on EAS attitude [15]. Fourth, although participants completed the self-reported questionnaire, the presence of the interviewer might not allow for privacy and anonymity. Fifth, with a response rate of only 55.6%, findings from this study’s sample population may not represent the general population due to non-response bias. Sixth, we could not perform inverse probability weighting techniques due to a lack of information of non-responders and, therefore, a concern of selection bias remains; the sociodemographic characteristics of the survey participants included in this study (n=1,000) were similar to those of the Korean population with regard to age (20–29 years: 15.3%, 30–39 years: 15.5%, 40–49 years: 18.9%, 50–59 years: 20.0%, 60–69 years: 16.7%, ≥70 years: 13.6% in the Korean population) and sex (men: 49.9%, women: 50.1% in the Korean population), suggesting low possibility of selection bias. Finally, we surveyed a small group of individuals aged over 70 and a large group of persons who were under 50.
Comment #8. Line 169: please correct “. [24[ Prior”.
We have corrected the error in Line 169.
Reviewer 2 Report
Attitudes toward the legalization of euthanasia
or physician-assisted suicide in South Korea (IJERPH 2022)
This manuscript describes a South Korean study of public attitudes toward the legalization of euthanasia or physician-assisted suicide (EAS) and of the reported reasons for the attitudes. A cross-sectional survey of a representative national sample of 1,000 people aged 19 years or older was administered in 2021. A finding was that seventy-six percent of the participants agreed with legalizing EAS in Korea. In the abstract it is stated that “participants that agreed with the legalization of EAS chose the “meaninglessness of the rest of life” and the “right to a good death” as their main reasons for their positive attitudes. Participants that disagreed with the legalization of EAS chose “respect for life,” “violation of the right to self-determination,” “risk of abuse or overuse,” and “violation of human rights” as their main reasons. . . . participants with poor physical status . . . or comorbidity . . . showed positive attitudes toward the legalization of EAS.”
A contribution of this paper is that the study was conducted in a country with limited research on EAS attitudes. A strength of its method is that representative national sample was used.
Comments and recommendations
Introduction:
The introduction missed a major systematic review of attitudes about EAS (Castelli Dransart et al., 2021). This systematic review should be included. Specifically, the findings of this review as well as their recommendations for research should be integrated in the introduction and in the discussion of the study.
Only one prior South Korean study is cited in the intro (Yun et al. 2018). This study should be described in full because it appears to be the single antecedent of this study. Any other EAS studies done in South Korea should also be included and fully described in the intro.
Method:
In this section it is stated: “The survey was conducted via individual face-to-face interviews. The interviewer visited the home or workplace of each individual selected for the sample to conduct an eligibility evaluation.”
The individual face-to-face interviews method was a way to evaluate eligibility--which is a strength of the study. A problem is that interviews do not allow for privacy and anonymity (despite what is stated on p. 3, that study's participants were granted anonymity). The data may be skewed in the direction of whatever response the participants believed that the interviewers desired. Therefore, at a minimum this study’s method should include information about the authors’ own views about the legalization of EAS and the authors’ basic demographic info (i.e., age and sex). The method should also include information about the data collectors’ basic demographic info (i.e., age and sex). Information about researchers’ demographics and positionality is routinely provided in qualitative interview-based research, in recognition of the influence that researchers views have on participants’ responses. This information therefore should be included in this manuscript as well.
The method/measure used for assessing EAS attitudes is described as follows:
“We constructed a questionnaire to examine attitudes toward the legalization of EAS and the reasons for these attitudes. Each participant’s level of agreement with the legalization of EAS was assessed with the following item: “In May of 2018, a 104-year-old Australian ecologist named Dr. David Goodall died by physician-assisted suicide in Switzerland. Two Koreans have already died from physician-assisted suicide, and 107 Koreans are members of a physician-assisted suicide group. The Netherlands, Luxembourg, Belgium, Switzerland, Canada, Australia, and eight other U.S. states allow euthanasia or physician-assisted suicide. In Korea, there is also an argument for legislating physician-assisted suicide or euthanasia. What do you think about physician-assisted suicide or euthanasia?” Responses were later assessed based on the following Likert scale: “very much agree” (1), “agree” (2), “disagree” (3), and “very much disagree” (4).
This in my view it the greatest weakness of the study. First, no justification is given for the choice of a new questionnaire, given that many exist in the literature. Second, the prompt used prior to the questions is exceedingly long. It also has too many layers--which means that there is no way to know what people responded to. Third, the prompt focuses on a man who is not Korean and who goes to Switzerland to die of PAS. In other words, the cultural context of the case prompt is completely foreign to the South Korea respondents. There is evidence that the context of EAS, that is how EAS questions are framed, makes a difference in how participant respond. Fourth, the case presented is that of a man. There is plenty of evidence sex of protagonist matters in attitudes toward EAS. In other words, the case prompt is anything but culturally appropriate and inclusive. Fifth, the attitude question asks about physician-assisted suicide or euthanasia when the case prompt is about physician assisted suicide. Sixth, the single item attitude measure is a problem. The authors need to explain these method choices and address their limitations in the discussion section.
It is stated that “data was collected on the reasons that the respondents agreed or disagreed with the legalization of EAS.” This is a strength of the study and also a weakness given that the interview method’s lack anonymity might have impacted the responses. The authors need to discuss this issue as a limitation of the study.
Results
Attitude data are reported in the aggregate. Given what is known about sex x age differences in attitudes about EAS, attitude data need to be reported by sex x age, in intersection.
Reasons data are also reported in the aggregate. Given what we know about sex and age differences in views of EAS reasons, data on reasons need to be disaggregated by sex and age, in interaction.
Discussion:
It is stated that “seventy-six percent of the participants agreed with legalizing EAS in Korea, which demonstrates substantial support from the Korean general public for EAS.” This statement/conclusion is not warranted based on the method (see above in terms of the fact that people were interviewed, that a problematic prompt was used to elicit the attitude response, and so on). Given the method serious problems (see above comment), the conclusions of this study need to be substantially restricted to what the problematic method allows to learn from this study.
It is unclear what is meant by the following statement: “our results support mandatory discussions of the legality of EAS in Korea as a safeguard for vulnerable populations.” How do the results (which have the major limitations noted above) call for “mandatory discussions of the legality of EAS in Korea as a safeguard for vulnerable populations?”
This statement “Previous studies have found that respect for autonomy and preferences for control over the end of one’s life are positively associated with support for PAS” misrepresents the complexity of positions argued in the literature--particularly the position of those who have been critical of legalization. I recommend that the authors access some of the main EAS legalization critiques, and include mention of them in the discussion. Some of the critiques appear in Castelli Dransart et al.’s article. Other important critical analyses of PAS legalization appear in publications by Meier 2020 (e.g., The slippery slope is real), Canetto 2019 (If physician assisted is a…. powerful choice…) and Foley & Hendin 2002 (The case against assisted suicide). Critical analyses of PAS legalization are particularly important for the discussion of this study’s results given that in this study “poor health and comorbidity …[were] associated with [positive] attitudes toward EAS.”
Author Response
Response to 2nd Reviewer
Comment #1. A major systematic review of attitudes about EAS (Castelli Dransart et al., 2021)
Thank you for your invaluable comments that prompted us to expand our idea about EAS. We have integrated the findings of major systematic reviews of attitudes about EAS (Castelli Dransart et al., 2021) as well as their recommendations for research in the INTRODUCTION and DISCUSSION of the manuscript as follows:
INTRODUCTION (page 2, lines 35-37)
Recent systemic reviews of older adults’ requests for or attitudes toward EAS showed that most studies investigated attitudes toward EAS, and that age, religiosity, education, and socio-economic status were consistent predictors of such attitudes [15].
DISCUSSION (pages 9, lines 186-190)
Recent systemic reviews of older adults’ attitudes toward EAS showed that younger age, higher education, higher socio-economic status, and lower religiosity were the most consistent predictors of attitude toward EAS [15]. However, findings from the reviews also indicated that there was difficulty in comparison across studies because of the variety of participants’ characteristics and outcome measures [15].
DISCUSSION (page 10, lines 243-244)
Future studies should adopt common and explicit definitions of EAS that allow for better consideration of personal, social, and cultural factors on EAS attitude [15].
Comment #2. Prior South Korean studies
We have added the description of two prior studies done in South Korea fully as follows (page 2, lines 52-57):
In 2008, 3,840 individuals—patients, family caregivers, general population, and physicians—from Korea showed that about 50% of those in the patient and general population groups supported EAS, as compared to less than 40% of family caregivers and less than 10% of physicians [19]. In 2016, four groups showed that about 30-40% of those in the patient and general population groups supported EAS, compared to about 20-30% of family caregivers and physicians who participated in the survey [6].
Comment #3. Individual face-to-face interviews
Survey participants responded to the self-reported questionnaire in the presence of the interviewer, who allowed for privacy and anonymity but could provide only further explanation of the study. As per the reviewer’s suggestion, we have changed the description of the interview format and added a more detailed explanation in METHODS and the limitation in DISCUSSION as follows:
MATERIALS AND METHOD (page 3, lines 72-78)
The survey was conducted in March-April, 2021. The interviewer visited the home or workplace of each individual selected for the sample to conduct an eligibility evaluation. We chose 1,800 eligible respondents to account for lower participation rates. Among them, 1,000 individuals who had a strong understanding of the survey’s purpose and method were finally recruited (response rate was 55.6%) and responded to the self-reported questionnaire in the presence of the interviewer. The interviewer alone could provide detailed explanations about the study and allow for privacy and anonymity.
DISCUSSION (page 10, lines 244-246)
Fourth, although participants completed the self-reported questionnaire, the presence of the interviewer might not allow for privacy and anonymity.
Comment #4. The method/measure used for assessing EAS attitudes
We have explained the method choices and addressed the limitations in the DISCUSSION as follows (pages 10, lines 236-244):
Third, the PAS of Dr. Goodall was reported in the media and became a big social issue in Korea, and this case was used in the survey to realistically understand EAS. As the prompt focuses on a man who is not Korean and who goes to Switzerland to die of PAS, the cultural context of the case prompt might be completely foreign to South Korean respondents. The context of EAS, that is, how EAS questions are framed, makes a difference in how participants respond. Additionally, a single-item attitude measure might not correctly gauge the attitudes of people toward EAS. Future studies should adopt common and explicit definitions of EAS that allow for better consideration of personal, social, and cultural factors on EAS attitude [15].
Comment #5. Lack of anonymity in the interview method
We added lack of anonymity in the interview method in limitation of DISCUSSION as follows (page 10, lines 244-246):
Fourth, although participants completed the self-reported questionnaire, the presence of the interviewer might not allow for privacy and anonymity.
Comment #6-1. Sex x age difference in attitude toward EAS
We appreciate the reviewer’s comments about expanding our idea about attitude toward EAS. We have reported attitude data on EAS by the intersection of sex x age in RESULTS with Table 2 as follows:
Table 2. Participants’ attitudes toward the legalization of euthanasia and physician-assisted suicide (n=1,000)
Male, N (%) |
|||||
Age |
20-49 |
≥50 |
|
||
Agree |
201(40.0) |
176 (35.0) |
|||
Disagree |
76 (15.1) |
50 (9.9) |
|||
|
Female, N (%) |
||||
Age |
20-49 |
≥50 |
|||
Agree |
200 (40.2) |
186 (37.4) |
|||
Disagree |
60 (12.1) |
51 (10.3) |
|||
RESULTS (page 4, line 128-129)
Table 2 shows the attitudes toward EAS according to the intersection of sex x age.
Comment #6-2. Sex x age difference in reasons toward EAS
We thank the reviewer’s suggestion to expand our idea about attitude toward EAS. As the reviewer suggested, we have reported the reasons for agreeing or disagreeing with the legalization of EAS according to the intersection of sex x age in Supplementary Table 1 as follows:
Table S1. Respondents’ reasons for agreeing or disagreeing with the legalization of EAS according to the intersection of sex x age
  |
Age (years), N (%) |
||||
Reasons for agreement (N = 763) |
Total N (%) |
Sex |
N (%) |
20-49 |
≥50 |
Meaninglessness of the rest of life |
235 (30.8) |
Male |
120 (31.8) |
55 (27.4) |
65 (36.9) |
Female |
115 (29.8) |
55 (27.4) |
60 (32.3) |
||
Right to a good death |
198 (26.0) |
Male |
107 (28.4) |
59 (29.4) |
48 (27.3) |
Female |
91 (23.6) |
50 (25.0) |
41 (22.0) |
||
Alleviation of suffering |
157 (20.6) |
Male |
67 (17.8) |
41 (20.4) |
26 (14.8) |
Female |
90 (23.3) |
51 (25.5) |
39 (21.0) |
||
Family suffering and burden |
113 (14.8) |
Male |
50 (12.9) |
25 (12.4) |
25 (14.2) |
Female |
63 (16.3) |
26 (13.0) |
37 (19.9) |
||
Social burden due to medical expenses and care |
35 (4.6) |
Male |
20 (5.3) |
13 (6.5) |
7 (4.0) |
Female |
15 (3.9) |
10 (5.0) |
5 (2.7) |
||
No violation of human rights |
24 (3.1) |
Male |
13 (3.4) |
8 (4.0) |
5 (2.8) |
Female |
11 (2.8) |
7 (3.5) |
4 (2.2) |
||
Importance of the right to self-determination |
1 (0.1) |
Male |
0 (0.0) |
0 (0.0) |
0 (0.0) |
Female |
1 (0.3) |
1 (0.5) |
0 (0.0) |
||
Reasons for disagreement (N = 237) |
Total N (%) |
Sex |
N (%) |
20-49 |
≥50 |
Respect for life |
105 (44.3) |
Male |
47 (37.3) |
28 (34.6) |
19 (42.2) |
Female |
58 (52.3) |
31 (50.0) |
27 (55.1) |
||
Violation of the right to self-determination |
37 (15.6) |
Male |
24 (19.0) |
14 (17.3) |
10 (22.2) |
Female |
13 (11.7) |
4 (6.5) |
9 (18.4) |
||
Risk of abuse or overuse |
31 (13.1) |
Male |
15 (11.9) |
9 (11.1) |
6 (13.3) |
Female |
16 (14.4) |
8 (12.9) |
8 (16.3) |
||
Violation of human rights |
29 (12.2) |
Male |
18 (14.3) |
13 (16.0) |
5 (11.1) |
Female |
11 (9.9) |
8 (12.9) |
3 (6.1) |
||
Risk of misdiagnosis |
23 (9.7) |
Male |
15 (13.5) |
12 (14.8) |
3 (6.7) |
Female |
8 (7.2) |
8 (12.9) |
0 (0) |
||
Possibility of recovery |
12 (5.1) |
Male |
7 (5.6) |
5 (6.2) |
2 (4.4) |
Female |
5 (4.5) |
3 (4.8) |
2 (4.1) |
EAS, euthanasia and physician-assisted suicide
Comment #7. Support from the Korean public for EAS.
We have changed the statement in DISCUSSION and CONCLUSION as follows:
DISCUSSION (page 8, lines 165-167)
Seventy-six percent of the participants agreed with legalizing EAS in the country, which indicates substantial growing support from the general public for EAS, similar to that found in countries such as England (75.8%) [10] and Switzerland (81.7%) [21].
CONCLUSION (page 10, lines 259-261)
Most of the survey participants agreed with the legalization of EAS, especially participants with poor physical status or comorbidity, suggesting that discussions about the legalization of EAS may commence in Korea in the near future.
Comment #8. Support for mandatory discussions of the legality of EAS.
We have changed the statement in DISCUSSION as follows (page 8, lines 168-169):
Our results suggest that a discussion about the legality of EAS may be necessary for Korea [25].
Comment #9. Critical analyses of EAS legalization.
We have added critical analyses of PAS legalization in DISCUSSION as follows (page 8-9, lines 218-227):
Although an EAS law might contain strict safeguards such as prognostic requirements and psychiatric exclusion, a broad definition of unbearable suffering, unpunished non-reporting, and expansion of EAS to children, people with mental illness, and dementia could limit the safeguards for vulnerable patients [42]. An individual autonomous choice for EAS is shaped by social norms and constrained by feasibility, and as a result, a slippery slope can be real [42,43]. Whereas unbearable suffering at the end of life is a common fear; people should know that they would be able to expect relief from suffering as a result of medical advances in palliative care [42]. Additionally, healthcare providers should be aware of the complexity of the EAS issue, and provide appropriate support and resources to patients in their decision-making process [15].

Reviewer 3 Report
I recommend the manuscript entitled" Attitudes toward the legalization of euthanasia or physician-assisted suicide in South Korea" for publication in IJERPH after a major revision.
The results reported in this paper are interesting. They regard the opinions of people who suport euthanasia and physician-assisted suicide and those who are against them.
The revision should address the points below, which substantially influence the interpretation of the results and the conclusions made in the article:
1.The authors surveyed a small group of individuals aged over 70 and a large group of persons who were under 50. This is a limitation that should be duly acknowledged in the Limitations section.
- The percentage of married participants is reported incorrectly. It is 71.4% and should be 7,41%,
3.If 7.14% of the participants were married and 28.6% were not married, what was the marital status of the remaining 60%? (might the results have been affected by the fact that the majority of the people surveyed were single?).
4.The authors use the term "participants’ depressive status ". How was this status determined? Depression is known to considerably affect a person’s decision to commit suicide and their desire to live.
5.It is unclear what the authors mean by „comorbidities” with regard to their sample. The participants represented a very broad age range, and the authors don’t say what particular comorbidities they had. These diseases might have been or might not have been relevant to the study reported in the manuscript.
6.I believe that the authors’ conclusion that " is a far-fetched generalization, given the remarks in 5.
The Conclusions need to be revised.
Author Response
Response to 3rd Reviewer
Comment #1. A small group of individuals aged over 70 and a large group of persons who were under 50
We have added this limitation in DISCUSSION of the study as follows (page 10, lines 255-256):
Finally, we surveyed a small group of individuals aged over 70 and a large group of persons who were under 50.
Comment #2 and #3. The percentage of married participants
We thank the reviewer for pointing out the mistake in the percentage of married participants. The correct number and percentage of married participants are 714 and 71.4% in Table 1. We have made the changes.
Comment #4. The term "participants’ depressive status "
We thank the reviewer for pointing out the mistake in using the term "participants’ depressive status ". We have corrected the sentence in RESULTS and made changes in Table 1.
RESULTS (page 4, lines 125-126)
Table 1 presents the participants’ sociodemographic characteristics, including their health status.
Comment #5. What particular comorbidities
We have added the particular comorbidities in Table 1.
Table 1. Participants’ sociodemographic characteristics, including their health status
Variable |
Description |
N |
% |
|
|
Mean |
SD |
Age |
|
47.96 |
14.66 |
|
|
N |
% |
Age |
20-29 |
166 |
16.60 |
|
30-39 |
166 |
16.60 |
|
40-49 |
205 |
20.50 |
|
50-59 |
209 |
20.90 |
|
60-69 |
164 |
16.40 |
|
≥70 |
90 |
9.00 |
Sex |
Male |
503 |
50.30 |
|
Female |
497 |
49.70 |
Cormobidity |
No |
734 |
73.40 |
|
Hypertension |
|
|
|
Dyslipidemia |
|
|
|
Diabetes mellitus |
70 |
7.00 |
|
Musculoskeletal disease |
24 |
2.40 |
|
Liver disease |
10 |
1.00 |
|
Others |
26 |
2.60 |
Education |
College graduate |
541 |
54.10 |
|
High school graduate |
361 |
36.10 |
|
Middle school or less |
90 |
9.80 |
Income |
≥5,000 |
276 |
27.60 |
4,000-5,000 |
274 |
27.50 |
|
|
3,000-4,000 |
228 |
22.80 |
|
<3.000 |
221 |
22.10 |
Marriage |
Married |
714 |
71.40 |
Not married |
286 |
28.60 |
|
Residence |
Urban |
460 |
46.00 |
|
Rural/suburban |
540 |
54.00 |
Religion |
Having religion |
360 |
36.00 |
|
No religion |
640 |
64.00 |
Job-status |
Occupied |
747 |
74.70 |
|
Non-occupied |
253 |
25.30 |
Comment #6. Our conclusion that is a far-fetched generalization
We have changed the CONCLUSION as follow (page 10, lines 259-261):
Most of the survey participants agreed with the legalization of EAS, especially participants with poor physical status or comorbidity, suggesting that discussions about the legalization of EAS may commence in Korea in the near future.
Round 2
Reviewer 1 Report
In my opinion a revision of English is needed.
Line 35: please change “Recent systemic reviews” in “Recent systematic reviews” (the same in line 137).
The power analysis is not correct. What does it mean = 40? Please, correct it.
Author Response
Response to 1st Reviewer
Comment #1. The revision of English
As requested by the academic reviewer, we requested the Editage to carefully review the grammar in our manuscript and have attached the editing certificate.
Comment #2. Change of “Recent systemic reviews” in “Recent systematic reviews
As requested by the academic reviewer, we change the word ‘systemic’ to the ‘systematic’ as follows:
INTRODUCTION (page 1, lines 36-38)
Recent systematic reviews of older adults’ requests for or attitudes toward EAS showed that most studies investigated attitudes toward EAS, and that age, religiosity, education, and socio-economic status were consistent predictors of such attitudes [15].
DISCUSSION (pages 8, lines 186-189)
Recent systematic reviews of older adults’ attitudes toward EAS showed that younger age, higher education, higher socio-economic status, and lower religiosity were the most consistent predictors of attitude toward EAS [15]. However, findings from the reviews also indicated that there was difficulty in comparison across studies because of the variety of participants’ characteristics and outcome measures [15]
Comment #3. The paragraph about sample size
As requested by the academic reviewer, we change the paragraph about sample size as follows (page 3, lines 105-108):
We used Gpower to set the appropriate sample size to keep the default setting for effect size (0.2), α (0.05), and 1-β (0.95). Considering gender (2, male, female), age (5, 20's, 30's, 40's, 50's, 60's or older), and regional size (4, special, wide area, city, county), the number of groups was the appropriate sample size was 1,000.